# The SUV4-20H Histone Methyltransferases in Health and Disease

**DOI:** 10.3390/ijms23094736

**Published:** 2022-04-25

**Authors:** Davide Gabellini, Simona Pedrotti

**Affiliations:** IRCCS San Raffaele Scientific Institute, Division of Genetics and Cell Biology, 20132 Milan, Italy; gabellini.davide@hsr.it

**Keywords:** SUV4-20H1, SUV4-20H2, H4K20, epigenetics, human diseases

## Abstract

The post-translational modification of histone tails is a dynamic process that provides chromatin with high plasticity. Histone modifications occur through the recruitment of nonhistone proteins to chromatin and have the potential to influence fundamental biological processes. Many recent studies have been directed at understanding the role of methylated lysine 20 of histone H4 (H4K20) in physiological and pathological processes. In this review, we will focus on the function and regulation of the histone methyltransferases SUV4-20H1 and SUV4-20H2, which catalyze the di- and tri-methylation of H4K20 at H4K20me2 and H4K20me3, respectively. We will highlight recent studies that have elucidated the functions of these enzymes in various biological processes, including DNA repair, cell cycle regulation, and DNA replication. We will also provide an overview of the pathological conditions associated with H4K20me2/3 misregulation as a result of mutations or the aberrant expression of SUV4-20H1 or SUV4-20H2. Finally, we will critically analyze the data supporting these functions and outline questions for future research.

## 1. Introduction

Post-translational methylation of lysine residues on histone tails is a key chromatin modification underpinning gene regulation and several cellular processes [1], and is dynamically mediated by specific histone lysine methyltransferases (KMTs) and demethylases (KDMs). The methylation of histone H4 was one of the first histone post-translational modifications to be discovered, nearly half a century ago. It is associated with many physiological processes, including heterochromatin formation [2,3,4], cell cycle regulation [5], DNA damage repair, and recombination [6]. In mammalian cells, the majority of histone H4 methylation is detected at the N-terminal tail on lysine 20 (H4K20) [7]. This methylation mark is evolutionarily conserved from yeast to humans [7] and exists in three distinct states: mono (H4K20me1)-, di (H4K20me2)-, or tri-methylation (H4K20me3). While H4K20me2 is the predominant H4K20 methylation state, found in 80% of total histone H4, H4K20me1 and H4K20me3 are less abundant and typically enriched in transcriptionally active and silent chromatin, respectively [8].

There is only one known H4K20 mono-methyltransferase, SET8 (KMT5A, also known as PR-SET7). However, there are several known di- and tri-methyltransferases, among which SUV4-20H1 (KMT5B) and SUV4-20H2 (KMT5C) are responsible for the vast majority of these two modifications [3,9]. H4K20me3 is typically a mark of silenced heterochromatin, which is also characterized by other histone modifications, such as H3K9me3 and H3K64me3, and is mainly found near chromosome centromeres and telomeres, where it ensures the correct segregation and integrity of the genome [10,11]. H4K20me3 and H3K9me3 have also been found to control imprinted gene expression by localizing to silenced imprinted gene promoters and non-expressed pseudogenes [12,13]. SUV4-20H1 and SUV4-20H2 are highly homologous and partially redundant, and have been investigated together in many studies [3,10]. Structurally, SUV4-20H1 and SUV4-20H2 share: (1) a catalytic SET domain, (2) a unique N-terminal domain compared to other SET domain-containing proteins, and (3) a Zn-binding post-SET domain [14]. Interestingly, while human *SUV4-20H2* and mouse *Suv4-20h2* genes produce relatively few transcript variants, *SUV4-20H1*/*Suv4-20h1* genes display an elaborate array of isoforms (Figure 1), suggesting that transcriptional and post-transcriptional regulation play an important role in controlling SUV4-20H1/Suv4-20h1 activity and/or localization in different tissues and developmental stages, or upon stimuli. Further studies are required to elucidate the physiological and pathological relevance of these aspects.

While PHF8 (KDM7B) and LSD1 (KDM1A) have been reported as H4K20me1 demethylases in mammals [16,17], two DNA repair proteins, hHR23A/B, have recently been demonstrated to be demethylases for H4K20me1/2/3 both in vivo and in vitro [18]. These two proteins, homologues of RAD23, are required for embryonic development and are able to demethylate H4K20me1/3 at their binding site. It is worth noting that both proteins had no activity on H4K20me2 under the same experimental settings, suggesting that hHR23A/B may not demethylate H4K20me2 under physiological conditions. Interestingly, hHR23A/B regulate the transcription of repetitive elements and mRNA by demethylating H4K20me3 and H4K20me1, respectively [18].

H4K20 methylation is essential for normal development, as implied by the finding that Suv4-20h double-null (dn) mice are perinatally lethal and their chromatin has nearly lost all H4K20me3 and H4K20me2, resulting in a genome-wide transition to H4K20me1 [10]. This condition is associated with increased genome stress sensitivity and defective DNA damage repair, revealing an important function for the Suv4-20h enzymes in developmentally programmed pathways for DNA rearrangements [10]. In addition to its role in ensuring chromosome segregation and genomic integrity [7], heterochromatin function is involved in various aspects of human health. Heterochromatin-induced gene silencing is important for mediating developmental transitions, and altered heterochromatic states can impair normal gene expression patterns, leading to the development of different diseases [19]. 

In this review, we will first discuss the physiological role of Suv4-20h and its associated epigenetic marks. Then, we will examine human diseases that are connected to aberrant H4K20 di- and tri-methylation states as a consequence of altered Suv4-20h1/h2 expression, such as aging, cancer, metabolic syndrome, and neuronal/muscular disorders.

## 2. Role of Suv4-20h in Health

### 2.1. Role in Replication and Cell Cycle Progression

The methylation of H4K20 is highly dynamic throughout the cell cycle (Figure 2). PR-Set7 and H4K20me1 are normally absent during DNA replication and peak during mitosis [20]. Conversely, Suv4-20h1/h2 and H4K20me2/me3 are at their highest levels during the G1 and S phases [20]. The misregulation of either or both PR-Set7 and Suv4-20h enzymes leads to increased sensitivity to DNA damage and defects in cell proliferation [20]. In somatic cells, DNA replication is initiated from origins of replication (ORIs) that are loaded with replication factors during mitosis and early G1. H4K20 methylation states are mandatory for proper DNA replication [8], and the direct involvement of H4 methyltransferases in the loading of replication origins is supported by several reports [20,21,22].

Pioneering work from Denny Reinberg’s group demonstrated that PR-Set7 recruits the origin recognition complex (ORC) to chromatin through Suv4-20h1/h2 [20]. Re-replication defects observed in cells overexpressing PR-Set7^PIPM2^, a non-degradable mutant of PR-Set7, were due to the overabundance of H4K20me2/me3 [20]. H4K20 methylation enhances the interaction between histone H4 and ORC in vitro, serving as docking sites for ORC1 and ORCA/LRWD1, which in turn recruit ORC to specific replication sites [20]. In particular, Long et al. demonstrated that the histone variant H2A.Z directly binds SUV4-20H1 in HeLa cells, promoting H4K20me2 deposition, which in turn is required for ORC1 binding and the proper activation of early replication origins [23]. Julien’s lab further demonstrated that the H4K20me state has broader roles in ensuring proper DNA replication and serves as an enhancer for MCM2-7 helicase loading and replication activation at defined origins [8]. In line with this, the authors revealed that Suv4-20h-mediated H4K20me3 stimulates the binding of ORCA and the pre-replication complex at a subset of late-firing origins, which is essential for timely replication during late S-phase. Indeed, the ability of Suv4-20h to enhance replication origin activity plays an essential role only at specific chromatin regions; in particular, it is required for the correct replication timing of late domains corresponding to heterochromatin regions [8].

The fluctuation in H4K20 methylation status is also tightly regulated during embryonic development, which requires a unique reprogramming mechanism to revert it to a ground epigenetic state and sustain a new developmental program [24]. In mice, after fertilization, only H4K20me1 is found throughout the preimplantation stages. Conversely, H4K20me2 is found at the 4-cell stage and becomes clearly detectable at the morula stage. H4K20me3 is completely absent in preimplantation embryos and is detectable only in the female chromatin of the zygote [24]. Heterochromatin starts becoming visible at the implantation stage and is detectable at E14.5 during post-implantation embryo development [25]. Hence, H4K20me3 is accumulated slowly throughout fetal development, suggesting that heterochromatin is associated with more differentiated cell lineages. Indeed, both of the heterochromatin markers H3K20me3 and HP1α disappear from the maternal genome before fertilization and do not reappear until late mid-gestation [25]. A lack of H4K20me3 may be required for zygotic reprogramming to take place. Indeed, the absence of H4K20me3 is correlated with the highest developmental potency [24]. The ectopic expression of Suv4-20h1/h2 results in embryonic developmental defects beyond the two-cell stage, indicating that H4K20me3 remodeling is required for developmental progression [24]. Suv4-20h2 ectopic expression causes replication stress and S-phase arrest, leading to a proliferation defect accompanied by replication abnormalities [24]. These findings shed light on the functional role of the absence of H4K20me3 during preimplantation development and suggest that, in contrast to somatic cells, H4K20me3 is incompatible with the fast progression of DNA replication in embryonic chromatin. In line with this, SUV4-20H2 was found in another study to regulate embryonic stem (ES) cell differentiation [26]. SUV4-20H2 knockout (KO) and SUV4-20H double KO ES cells displayed a complete loss of H4K20me3, which resulted in altered gene expression and delayed differentiation [26]. RNA-sequencing (RNA-seq) revealed that SUV4-20H enzymes repress lineage-specific genes in ES cells [26]. Further work is needed to elucidate the role of H4K20me3 in human embryonic development.

The concept that heterochromatin silencing has a role in maintaining cell-lineage-specific epigenetic patterns of gene expression is well recapitulated during mouse epidermal development, where H4K20me3 is maintained at low levels in basal progenitor cells with higher epigenomic plasticity, while it is enriched upon terminal differentiation into suprabasal cells to bring about gene silencing [27]. 

### 2.2. Role in Recombination

The proper regulation of H4K20 methylation throughout the cell cycle is of great importance to preserve cellular homeostasis (Figure 2). The methylation of H4K20 plays a pivotal role in maintaining telomere length homeostasis in mammalian cells. Indeed, the loss of heterochromatic features in telomeric and subtelomeric chromatin have been shown to result in telomere length deregulation [28]. Cells deficient for Suv4-20h2 or for both Suv4-20h1 and Suv4-20h2 show decreased levels of H4K20me3 at telomeres and subtelomeres, leading to telomere elongation and increased frequencies of telomere recombination in the absence of changes in subtelomeric DNA methylation [2].

Suv4-20h enzymes also function in immunoglobulin class-switch recombination (CSR), in concert with the activation-induced cytidine deaminase gene AID [6,10]. CSR in the immunoglobulin heavy chain (IgH) locus is a process that changes the antibody effector function by replacing the default constant region (Cμ) of the antibody gene with a different constant region [6]. This process is not sequence specific and involves the generation of DSBs in highly repetitive switch (S) regions, which precede every individual constant region. CSR is initiated by the AID-induced deamination of cytosines to uracils, leading to the generation of dU:dG mismatches that are differentially processed to generate double-strand breaks in Ig switch regions in CSR [6]. AID activity is primarily controlled through tissue-specific and stage-specific expression upon cell activation. Its transcript is regulated by both ubiquitous and lymphoid-specific transcription factors (Pax-5, STAT6, SP1, and C/EBP) and miRNAs (miR155 and miR181b) [29]. Recently, it was reported that AID is required for recruiting Suv4-20h enzymes to CSR sites by physically interacting with the enzymes [6]. This evidence also supports previous reports showing defective CSR in Suv4-20h-dn B cells [10]. While these data demonstrate an important function for Suv4-20h enzymes to ensure the lineage program of DNA-rearranging lymphoid cells, further work is required to elucidate how their interaction with AID is regulated.

### 2.3. Role in Transcriptional Regulation

In addition to their well-recognized role in constitutive heterochromatin maintenance, H4K20me3 and Suv4-20h proteins are active modulators of gene expression (Figure 2). In embryonic stem (ES) cells, SUV420H2 and H4K20me3 control chromatin architecture and are enriched at DNA sequences containing repetitive elements, particularly endogenous retroviruses (ERVs) [26]. It has been hypothesized that the SUV420H2-dependent silencing of ERVs in ES cells could prevent their activation during early development. Surprisingly, in ES cells, H4K20me3 has been found to interact with many transcripts from loci that are actively transcribed, suggesting that H4K20me3 marks transcriptionally dynamic regions in ES cells; this is supported by the observation that the expression of H4K20me3-associated RNAs is predominantly enriched in undifferentiated ES cells compared to differentiated cells [30,31]. Numerous mRNA, ncRNA, and proteins have been found to associate with H4K20me3, which may contribute to the regulation of genome structure and expression [30,31]. In one study, chromatin-associated RNA immunoprecipitation followed by sequencing (CARIP-Seq) [30] revealed that, preferentially, H4K20me3 interacts with longer RNA transcripts with a greater number of exons, as has also been reported for many other histone-modifying enzymes, such as PRC2 and HDAC1.

Suv4-20h1/h2 and H4K20me3 have also been found to play an evolutionarily conserved role in the direct regulation of Oct4-related genes [32] as well as RNA polymerase II promoter proximal “pausing” by antagonizing histone acetyltransferase hMOF-mediated acetylation of H4K16 at human CpG-island genes [33].

Examples of Suv4-20h1 regulated transcription come from work performed by Chinenov et al., who found that Suv4-20h1 was able to interact with the nuclear receptor GRIP1 to negatively regulate GR-dependent transcriptional activation in mammalian cells [34]. GRIP1 belongs to the p160 family of nuclear receptor coactivators, which regulate transcription in a hormone-dependent fashion. Chinenov et al. demonstrated that Suv4-20h2 interacts with the GRIP1 repression domain (RD) and antagonizes the glucocorticoid-dependent activation of GR target genes.

The trimethylation of H4K20 is dynamically regulated during cell growth, proliferation, and differentiation. Upon growth arrest because of cell density or serum deprivation, H4K20me3 increases to establish chromatin compaction at repetitive sequences. Conversely, upon mitogenic stimuli, there is a substantial decrease in H4K20me3, which allows cells to re-enter the cell cycle. In growth-arrested cells, Suv4-20h2 interacts with quiescent-induced lncRNA PAPS (promoter and pre-rRNA antisense) to trigger the transcriptional silencing of rRNA genes (rDNA) [35]. It was recently demonstrated that Suv4-20h2 is recruited by the epigenetic regulator plant homeodomain (PHD) finger protein 6 (PHF6) to silence rDNA transcription [36]. Under physiological conditions, gene body methylation is largely maintained by DNA methyltransferase 1 (DNMT1), which ensures the proper transcription of rRNA genes [36]. In response to developmental and/or environmental stress, PHF6 binds hypomethylated rDNA gene bodies because of DNA methyltransferase 1 (DNMT1) deficiency or downregulation, where it engages with Suv4-20h2 to establish H4K20me3 and suppress rDNA transcription [36]. In addition, a critical role of H4K20 methylation—in particular, H4K20me3—in regulating the transition between quiescence and proliferation has been reported [37]. The consistent upregulation of H3K20me3 in multiple different quiescent models indicates it may play a larger role in establishing the functional state of chromatin in quiescence and may regulate specific gene expression changes necessary for quiescence entry, exit, and maintenance.

H4K20 methylation marks can also cooperate with other histone marks. For instance, H4K20me3 has been detected as part of bivalent promoters in combination with H3K4me3 and H3K36me3 [38], and as part of heterochromatin in combination with H3K9me3 [3]. Taken together, these data support the possibility that H4K20me3 represents one part of a combinatorial code that regulates quiescence.

## 3. Suv4-20h in Disease

### 3.1. Aging

Cellular senescence is a well-orchestrated and programmed process involved in embryonic development, physiological aging, age-related pathologies, and cancer [39]. Cellular senescence is associated with an altered pro-inflammatory secretory pathway and an important tumor suppressor mechanism [40]. Senescent cells acquire distinctive features, including a stable cell cycle arrest, senescence-associated β-galactosidase (SA-β-gal) activity, and marked alterations in higher-order chromatin organization, that are associated with significant changes in gene expression [41,42].

Consistent with heterochromatin formation during senescence, a marked increase in H4K20me3 has been reported in senescent cells, both oncogene-induced as well as in replicative senescence [41], progeroid cells [43], and physiologically aged tissues [44] (Figure 2). However, whether the increase in H4K20me3 and the enzymes responsible for its deposition, Suv420h, are required for the onset rather than the maintenance of the senescent phenotype has not been clearly addressed. An analysis of the genomic distribution and function of H4K20me3 in senescent cells from Peter Adams’ lab demonstrated that an increase in H4K20me3 during senescence, although not sufficient to trigger senescence, helps to stabilize the senescent epigenome and genome and, consequently, maintain the senescent phenotype [42]. In support of this notion, in senescent cells, H4H20me3 is enriched at genes that are already lowly expressed or unexpressed, suggesting that this epigenetic modification corresponds to the maintenance of the repressed epigenome. Interestingly, this work showed that augmented H4K20me3 in senescent cells was not accompanied by the increased expression of Suv4-20h2 [42], leaving an open question about the mechanism(s) responsible. Furthermore, upregulated levels of H4K20me3 also represent a hallmark of aging diseases, such as Hutchinson–Gilford progeria syndrome (HGPS) [43]. Late-passage HGPS cells display an altered histone methylation status as a direct consequence of the expression of mutant lamin A (LAD50), leading to the perturbed epigenetic control of chromatin structure. Nevertheless, specific target senescence genes directly regulated by H4K20me3 and/or Suv4-20h proteins have not been identified yet. The discovery of direct targets would be of great interest for designing ad hoc interventions to prevent/ameliorate age-related diseases.

A progressive increase in the expression levels of both Suv4-20h1 and Suv4-20h2 has been reported during physiological ovarian aging in mice [45]. However, whether this increase is accompanied by augmented H4K20me3 levels is not known. Ovarian aging can lead to severe menopause symptoms, which affect women’s physical and mental health [45]; the identification of physiological and pathological regulatory mechanisms and molecular targets is of great interest for designing specific interventions. Hence, the relevance of Suv4-20h enzymes during the ovarian aging process is worthy of further exploration. 

While increased levels of H4K20me3 seems to be a common feature of senescent cells, Lyu et al. described the downregulation of H4K20me3 as a consequence of TGF-β activation triggered by environmental stress and endogenous signals, such as hyperoxia-induced oxidative stress, which is known to activate TGF-β pathway [39,46]. TGF-β/Smad signaling is a well-known pathway that regulates both damage-induced senescence and developmentally programmed senescence [47]. Lyu and coworkers demonstrated that upon induction, TGF-β/Smad signaling activates mir-29 expression, which in turn mediates the loss of H4K20me3 through the direct repression of its catalyzing enzymes Suv4-20h1 and Suv4-20h2. Considering that previous reports showed a marked upregulation of H4K20me3 and Suv4-20h proteins during senescence [41,42,43,44], the authors speculated that their paradoxical result could be due to cell type-specific and context-dependent TGF-β signaling. Indeed, they showed the restoration of H4K20me3 abundance in the senescent heart, but not in the aging kidney, lung, and spleen, following the abolition of TGF-β signaling which inhibited miR-29 expression, corroborating the hypothesis that the cell and tissue contexts determine the specific response to TGF-β during senescence and aging. 

The context-dependent activity of H4K20me3 has also emerged in cancer, highlighting the complexity of such polyhedric epigenetic modifications. In the next section, we will discuss how Suv4-20h enzymes and H4K20me3 have been implicated both in tumorigenesis and tumor suppression.

### 3.2. Cancer

Aberrant histone modifications are a hallmark of cancer [48]. During cell reprogramming, the loss of H4K20me3 leads to enhanced telomere elongation, generating cells with a faster teratoma growth rate and providing a growth advantage and increased tumorigenesis potential to the tumor cells [49] (Figure 2). These results indicate that the abrogation of Suv4-20h enzymes and the loss of the heterochromatic mark H4K20me3 in telomeric heterochromatin facilitates telomere reprogramming and provides an increased tumorigenic potential. Accordingly, the genome-wide loss of H4K20me3 is observed in multiple types of cancer, and it is correlated with poor prognosis [48,50]. In addition, at intracisternal A-particle (IAP) sequences, chromatin compaction via the tri-methylation of H4K20 likely impairs retrotransposition in postmitotic cells that cannot employ high-fidelity homologous recombination between sister chromatids [35]. This finding becomes highly relevant in the cancer field, implying that high levels of H4K20me3 in somatic cells are important to counteract cancer-associated somatic transposition, providing one possible molecular explanation for why H4K20me3 is progressively lost in human tumors.

A recent analysis of COSMIC mutational data revealed that Suv4-20h1 is frequently mutated in cancer cells [51], having the seventh highest fraction of somatic tumor mutations among all KMTs. Several of these somatic mutations lead to a reduction in its catalytic activity, suggesting that Suv4-20h1 might function as a tumor suppressor gene [51]. Mutations in the SUV4-20H1 gene have also been associated with acute myeloid leukemia (AML) pathogenesis [52]. Although genome-wide association studies have identified SUV4-20H1 as a major susceptibility locus for AML, how this contributes to AML onset and progression is currently unclear. SUV4-20H1 inactivating mutations have also been reported in pediatric glioblastoma tumors. Despite the fact that they are present in <1% of cells, these mutations abrogate DNA repair and confer increased invasion and migration on neighboring cells, in vitro and in vivo, through chemokine signaling and the modulation of integrins [53]. The SUV4-20H1 gene also displays an altered DNA methylation and hydroxymethylation profile in glioblastoma multiforme (GBM), the most common and aggressive type of brain tumor in adulthood [54]. Hypermethylation and hypo-hydroxymethylation of the SUV4-20H1 gene correlate with its decreased expression in some GBD specimens compared to normal brain tissue, suggesting that SUV4-20H1 downregulation is relevant at least for a subset of GBM tumors. Accordingly, the restoration of SUV4-20H1 expression, and consequently H4K20me2 levels, reduces tumor growth in vivo in mouse xenografts [54]. 

Human breast cancer cells are characterized by prominent epigenetic alterations that are associated with the increased malignant properties of cancer cells [55]. The loss of H4K20me3 has been observed in animal models of breast carcinogenesis, and H4K20me3 levels are reduced in malignant breast cancer-derived cell lines relative to those in nontumorigenic breast epithelial cells [56]. Reduced H4K20me3 in tumor cells is associated with advanced stages of breast cancer and poor prognosis. Accordingly, SUV4-20H2 overexpression in the metastatic MDA-MB-231 breast cancer cell line suppresses cell invasion at least in part by the repression of cancer promoting genes, among which is tensin-3, an Src substrate known to regulate cell migration [57,58]. 

miRNAs represent an additional layer in the regulation of tumor progression. Cancer cells exhibit distinct miRNA expression profiles, which can contribute to carcinogenesis. Breast cancer stem cells (BCSCs) have been demonstrated to express a distinct miRNA expression profile compared to other breast cancer cells [59,60]. miR29a has been found to be upregulated in BCSCs, as well as in aggressive breast cancer cell lines and breast cancer tissues [60]. By suppressing the expression of SUV4-20H2, which leads to the loss of H4K20me3, miR-29a attenuates the repression of CTGF and EGR1, thus promoting the epithelial-to-mesenchymal (EMT) progression and metastasis of breast cancer cells. miR29a can be induced by bFGF, a growth factor secreted by tumor microvascular endothelial cells (tMVECs) that has been reported to enhance the invasive potential of cancer cells. Although miR29a is upregulated and functions as an oncogene in breast cancer and BCSCs, it has been identified as a tumor suppressor gene in other types of cancer, being downregulated in gastric cancer, pancreatic cancer, and prostate cancer [61]. These conflicting reports regarding the role of miR29a and the potential interplay with SUV4-20H2 in different types of cancer deserve further research.

A correlation between H4K20me3 levels and tumor development has also been reported in human lung tumors [50,62]. Most lung cancer patients are diagnosed with non-small cell lung cancer (NSCLC), a subtype that represents 85% of lung cancer cases. The TCGA and the Genotype-Tissue Expression (GTEx) projects revealed lower SUV4-20H2 levels in both lung adenocarcinoma (LUAD) and lung squamous cell carcinoma (LUSC) samples relative to normal samples, suggesting that SUV420H2 functions as a bona fide tumor suppressor. In a very recent paper from Pal et al., the loss of SUV4-20H2 was associated with acquired erlotinib resistance in NSCLC, identifying a pivotal role for SUV420H2 in mediating drug resistance [62]. Indeed, the SUV4-20H2 expression level was negatively correlated with erlotinib response in a panel of NSCLC cell lines. NSCLC cells with downregulated SUV4-20H2 express high levels of the oncogenic long non-coding RNA LINC01510, which transcriptionally upregulates the oncogene MET. Increased levels of MET lead to erlotinib (an epidermal growth factor receptor inhibitor used as standard-of-care treatment for NSCLC patients harboring EGFR alterations) resistance. Interestingly, EGFR has been demonstrated to interact with and phosphorylate histone H4 at the Y72 residue [63], which facilitates the recruitment of both SET8 and SUV4-20H methyltransferases. This interaction leads to the methylation of H4K20 to promote DNA synthesis and repair, providing a new molecular mechanism for EGFR nuclear functions [63]. 

The maintenance of the malignant phenotype is often dependent on the initiating oncogenes. The MYC oncogene is overexpressed in more than half of human cancers and coordinates the expression of thousands of genes that could potentially contribute to its neoplastic properties. Work from Li et al. demonstrated that MYC maintains a neoplastic state through the regulation of the microRNA cluster miR-17-92, which controls specific chromatin regulatory and survival programs [64]. By using a conditional system, they screened for genes regulated by MYC and/or miR-17-92. Among the genes regulated by both MYC and miR17-92, they found chromatin modifiers that have not been previously reported as MYC targets, namely Sin3b, Hbp1, Btg1, and Suv4-20h1. MYC, through miR-17-92, directly suppresses the expression of the chromatin regulatory genes Sin3b, Hbp1, Suv420h1, and Btg1 and the proapoptotic gene Bim, which is causally required to maintain survival, autonomous proliferation, and self-renewal. 

While the loss of H4K20me3 has been regarded as a potential hallmark of human cancer, several reports describe paradoxically increased Suv4-20h2 and Suv4-20h1 expression in tumors [65,66,67,68]. In pancreatic cancer, there is a gradual increase in Suv4-20h2 expression as the disease progresses, partially due to gene amplification [68]. Suv4-20h2 acts as an epigenetic regulator of the EMT transition program by silencing the expression of key mesenchymal-to epithelial (MET)-promoting transcription factors via H4K20me3 [68]. A recent study from a meta-analysis identified that Suv4-20h1 is among the eight (out of fifty) dysregulated histone lysine methyltransferases in breast cancer induced by genetic alterations [66]. A gain or amplification of Suv4-20h1 has been observed in most breast cancer cell lines, where Suv4-20h1 is highly expressed compared with non-tumorigenic breast epithelial cells [66]. In addition, studies in the TCGA database showed that both Suv4-20h1 and Suv4-20h2 are amplified in many types of human cancers, including breast, esophageal, bladder, head and neck, colon, kidney, lung, liver, prostate, stomach, uterus, etc. [67,68]. The expression profiles of 11 squamous cell carcinoma cell lines have demonstrated that Suv4-20h1 is significantly highly expressed in these cells compared with normal keratinocytes [67]. Moreover, in leukemia K562 cells, the knockdown of Suv4-20h1 resulted in growth inhibition through the induction of G1 arrest during the cell cycle [69]. In this study, the authors demonstrated that Suv4-20h1 represses p21 expression and consequently prevents the inhibition of cyclins at the G1 checkpoint, thus promoting the G1/S transition.

There are several explanations for the results described above. During cell cycle progression, H4K20 methylation is tightly regulated by Set8/PR-Set7, Suv4-20h1, and Suv4-20h2, and changes in the expression level of one enzyme may not alter the cellular and/or local gene histone H4K20 methylation levels in different contexts, resulting in balanced H4K20 methylation, which allows cell cycle progression [69]. Additionally, some discrepancy may be due to differences in the genetic background or progressive stages of the cancer cells or specimens. For example, H4K20me3 is significantly altered in breast cancer cell lines and is correlated with the carcinogenic potential of the cells [55]. It is less pronounced in malignant MDA-MB-231 compared to lowly-invasive MCF-7 cells, while non-tumorigenic human MCF-10-2A epithelial breast cells display the highest levels. Accordingly, SUV4-20H2 expression is high in MCF-10-2A and progressively decreases in more aggressive MDA-MB-231 cells [55]. In line with results in breast cancer cell lines, Isin and coworkers demonstrated that SUV4-20H2 expression also decreases with breast cancer progression in breast tumor tissue [70]. Furthermore, Suv4-20h1 can target other proteins for methylation/interaction, an effect that may later play a role in the cell cycle. For example, Vougiouklakis et al. reported that Suv4-20h1 enhances the phosphorylation and transcription of ERK1 in cancer cells, thereby promoting cancer cell proliferation [67]. However, it is worth noting that work from Albert Jeltsch’s lab, who investigated the specificity of the Suv4-20h protein methylation activity of novel substrates, showed no activity of SUV4–20H proteins on ERK1 peptide substrates containing the target lysine residues K302 and K361 or on the ERK1 protein [71]. The authors pointed out the lack of methods describing the strategy used for SUV4-20H1 protein purification in Vougiouklakis’ paper. It is tempting to speculate that the observed methylation activity could be due to contaminant KMT(s) co-purified with SUV4-20H1. Finally, although the levels of SUV420H enzymes may increase in cancer, the concomitant high expression of demethylases might outcompete the methyltransferase activity, resulting in overall lower H4K20me3 levels. Careful analyses are required to determine the relationship between Suv4-20h expression and histone H4K20 methylation status in different contexts [69].

### 3.3. Metabolic Disorders

Metabolic diseases, such as obesity, metabolic syndrome, type 2 diabetes (T2D), and cardiovascular disease (CVD), are complex multifactorial disorders caused by genetic and environmental factors [72,73]. A growing body of literature suggests that epigenetic mechanisms play a crucial role in the development of metabolic disorders [72]. Aging and lifestyle factors, including diet and physical activity, have been demonstrated to have long-lasting effects on the epigenome [72]. The increasing number of people affected worldwide makes metabolic diseases a major threat to global health [72,73]. The systemic perturbation of organismal metabolism induced by obesity is also a major cancer risk factor, being associated to at least 13 types of cancer [74]. Several genes are activated or reduced to regulate energy metabolism, and epigenetic factors are the main mechanisms for modulating gene expression [75]. Hence, there has recently been extraordinary interest in the roles of epigenetic modifications in regulating energy metabolism.

Studies focused on epigenetic marks in obesity have found altered methylation and/or histone acetylation levels in genes involved in both specific and more general metabolic processes [76]. Histone modifications play key roles in regulating metabolic genes in response to environmental cues. Several studies have shown that Suv420h enzymes are regulated by environmental stimuli and play a role in metabolism regulation, suggesting an evolutionarily conserved role in obesity [1,77,78,79,80,81,82,83,84] (Figure 2). The double knockout of Suv4-20h1 and Suv4-20h2 mesenchymal precursor cells using Myf5-Cre increases brown adipose tissue (BAT) metabolic activity and enhances the browning of white adipose tissue (WAT), resulting in improved metabolic parameters and systemic protection against obesity [77]. The activation of the key metabolism regulating transcription factor peroxisome proliferator-activated receptor gamma (Ppar-γ) was reported to be the main mechanism mediating the improved metabolic phenotype in the Suv4-20h double-knockout mice [77]. Intriguingly, a more recent study showed that Suv4-20h2, but not Suv4-20h1, is required for thermogenic gene expression in adipocytes [85]. Knockout mice for Suv4-20h2 in adipocytes driven by Adipoq-Cre exhibit reduced adipose thermogenesis and are susceptible to DIO. The deletion of the Suv4-20h2 gene results in increased p53 expression and the activation of thermogenic genes [85]. The apparent discrepancy in the role of Suv4-20h2 in regulating adipose tissue function may result from different Cre lines (Myf5 versus adiponectin) that delete the gene in different tissues (BAT vs. WAT) or at different developmental stages (precursor vs. mature adipocytes). The H4K20me3 code may modulate distinct target genes in precursor and mature adipocytes, leading to diverse metabolic consequences. Hence, further studies are needed to better define the role of Suv4-20h in adipose tissue development and function. 

### 3.4. Neurodevelopmental Disorders

Neurodevelopmental disorders (NDDs) are defined as a heterogeneous diagnosis comprising various disabilities [86]. The targeted sequencing of candidate genes for NDDs has identified SUV4-20H1 as a high-risk gene harboring de novo mutations in NDD patients, including autism spectrum disorder (ASD) and intellectual disability (ID) patients [86,87] (Figure 2). In particular, variants causing SUV4-20H1 haploinsufficiency are associated with dominant developmental disorders [1]. Very little is known about the physiological role of SUV4-20H1 in the brain. Suv4-20h1 haploinsufficiency in mice results in global developmental delay and other ASD comorbidities (obsessive compulsion, depression, and anxiety) [87]. In this study, several outcomes differed by sex, perhaps mirroring the sex bias in ASD [87]. Suv4-20h1 expression is enriched in the prefrontal cortex (PFC), a brain region strongly implicated in autism [88]. Accordingly, PFC-specific Suv4-20h1 knockdown induces autism-like social deficits, which are linked to glutamatergic synaptic deficits in the PFC, a physiological phenotype commonly found in mouse models of autism [88]. Mechanistically, Suv4-20h1 deficiency leads to the accumulation of DNA double-strand breaks (DSBs) and increased p53 expression, highlighting the critical role played by Suv4-20h1 in DNA repair [88]. Moreover, Suv4-20h1 downregulation induces the activation of genes involved in cellular stress, resulting in the impairment of glutamatergic transmission and social deficits [88]. In line with these findings, single cell RNA-sequencing (scRNA-Seq) performed on organoid models of the human cerebral cortex confirmed that SUV4-20H1 haploinsufficiency confers the asynchronous development of two main cortical neuronal lineages—γ-aminobutyric acid-releasing (GABAergic) neurons and deep-layer excitatory projection neurons [89].

Previous studies have suggested a role for the enzyme in the cell cycle regulation of neural stem progenitor cells (NSPCs) [90]. In the adult brain, NSPCs in the subventricular zone (SVZ) are required for maintaining a fine balance between self-renewal and differentiation [90]. The disruption of such delicate equilibrium may shift normal neurogenesis to oncogenic transformation or neurodegeneration [90]. In this context, the Suv4-20h-associated H4K20me3 mark acts in synergy with EZH2/H3K27me3 to reduce improper gene expression and regulate cellular proliferation, preventing adult NSPCs from aberrant cell cycle re-entry or differentiation [90]. The pivotal role played by the Suv4-20h epigenetic regulators in the developing brain has been corroborated by subsequent studies in Drosophila that support a role for Suv4-20h in habituation learning (a form of non-associative learning in which an innate response to a stimulus decreases after repeated or prolonged presentations of that stimulus [91]) [86]. The aberrant expression of Suv4-20h proteins has also been described in neuromuscular disorders such as Friedreich’s ataxia (FRDA), which is caused by the decreased expression of the frataxin gene (FXN) [92]. Disease-associated GAA trinucleotide expansion is the leading cause of the partial silencing of the FXN locus. The mechanism driving GAA-mediated downregulation of the FXN gene has been elucidated very recently by Vilema-Enriquez and colleagues, who found H4K20me3 to be enriched in the flanking regions of the GAA repeats [92]. The downregulation of Suv4-20h expression by siRNA or the pharmacological inhibition of its catalytic activity was shown to restore FXN expression in primary FRDA patient-derived cells, identifying Suv4-20h1 as a potential therapeutic target for the disease.

### 3.5. Muscle-Related Disorders

The central role played by Suv4-20h and its associated H4K20 modification in regulating cell differentiation has also emerged during muscle cell differentiation. Quiescent muscle satellite cells (MuSCs), characterized by a high content of heterochromatin, display a discrete amount of H4K20me3, which decreases dramatically upon activation and proliferation [93] (Figure 2). This switch in constitutive heterochromatin content between quiescent and activated MuSCs is also recapitulated by the different distribution of Suv4-20h enzymes. Accordingly, while Suv4-20h1 is mainly present in quiescent MuSCs, Suv4-20h2 becomes enriched in differentiated cells. The depletion of Suv4-20h1 in MuSCs boosts stem cells proliferation, while the inactivation of Suv4-20h2 exerts the opposite effect, suggesting distinct roles of the two enzymes in the control of MuSC quiescence and proliferation [93]. The discovery of Suv4-20h1-mediated chromatin compaction and the regulation of quiescence in MuSCs paves the way for a deeper understanding of how MuSCs are replenished after injury and return to quiescence, thus opening exciting perspectives for therapeutic approaches.

Although Suv4-20h1 is dispensable for normal muscle development, the reduction in condensed heterochromatin upon the inactivation of Suv4-20h1 affects long-term skeletal muscle regeneration capacity, indicating that Suv4-20h1 is required for maintaining the quiescence of MuSCs, thereby enabling muscle regeneration [93]. Accordingly, the aberrant expression and/or cell localization of Suv4-20h1 has been associated with muscle disorders, as in the case of facioscapulohumeral muscular dystrophy (FSHD) [94,95]. Yeast two hybrid screening has identified Suv4-20h1 as an interaction partner of FSHD region gene 1 (FRG1). FRG1 overexpression in C2C12 muscle cells causes Suv4-20h1 mislocalization, interfering with its function. Accordingly, Suv4-20h1 knockdown in myoblasts recapitulates the differentiation defects observed upon FRG1 overexpression. One way Suv4-20h1 may regulate muscle cell differentiation is by silencing the inhibitor of differentiation gene Eid3 [95]. Hence, FRG1 overexpression, as reported in FSHD, could lead to muscle defects by sequestering Suv4-20h1 away from its epigenetic targets, leading to the derepression of genes inhibiting muscle cell differentiation [95].

## 4. Discussion

The methylation of H4K20 is essential for several cellular processes, and the aberrant regulation of H4K20 methylation states is linked to a variety of diseases, including cancer and developmental disorders (Figure 2). During recent years, there has been considerable progress in our understanding of the roles of histone H4 methylation and the catalyzing enzymes. H4K20me states are highly cell cycle-regulated, where mono- and trimethylation fluctuate the most. This tight regulation is crucial for embryonic development and cell differentiation programs, as the disruption of the dynamic fluctuations in H4K20 methylation results in developmental defects and several diseases. 

SUV4-20H protein upregulation is linked to several diseases, such as aging disorders, neuromuscular diseases, and various types of cancer. Hence, the development of SUV4-20H inhibitors holds great promise for disease treatment. The highly selective A-196 SUV4-20H inhibitor [96] has been shown to increase *FXN* expression in FRDA patient-derived cells [92], supporting the relevance of SUV4-20H inhibition for the treatment of neurological disorders associated with increased SUV4-20H expression or activity. One of the main concerns with epigenetic-based therapies is the potential broad biological effects that can result from the global inhibition of an epigenetic target. Schotta et al. reported that single *Suv4-20h1* or double *Suv4-20h1*/*Suv4-20h2*-null mice were perinatally lethal [10]. However, it is currently not known whether the ablation of Suv4-20h enzymes in adults causes any defects. Hence, to reduce side effects and toxicity, SUV4-20H inhibition should be modulated in time and space. In this regard, one possibility may come from nanotechnologies. The development of targeted nanosystems that make use of drug-loaded nanoparticles could be employed for delivering SUV4-20H inhibitors in a tissue-specific manner, thus avoiding toxic systemic effects.

Several open questions remain about the physiological and pathological role of Suv4-20h proteins. While *Suv4-20h1* is ubiquitously expressed during embryogenesis and in adult tissues, *Suv4-20h2* is much less abundant in the embryo, and its expression is restricted to few adult tissues (liver, testis, kidney, placenta, spleen, and thymus) [10]. Hence, they most likely play non-overlapping roles, which need to be elucidated. Although Suv4-20h1 is primarily responsible for H4K20me2, H4K20me3 is not completely lost in embryonic tissues of *Suv4-20h2^−/−^* mice, suggesting a compensatory function of Suv4-20h1. Furthermore, the gene regulatory activity of Suv4-20h proteins has never been assessed mechanistically, leaving a significant gap in the knowledge about how they regulate specific genes and genomic regions. Another piece of missing information concerns how Suv4-20h expression and activity are regulated. These represent crucial aspects, since they can help in designing ad hoc therapeutic interventions aimed at modulating Suv4-20h expression and/or activity. This would also require the development of novel techniques aimed at investigating the endogenous enzymes in a variety of assays. The relatively low expression level of the two enzymes and their tight association to heterochromatin has strongly affected the identification of direct targets that can mediate the biological effects of Suv4-20h. In this regard, one of the biggest drawbacks is the lack of specific antibodies for both Suv4-20h proteins, which are essential for applications such as ChIP-seq and/or immunofluorescence, impacting the identification of additional modifications or proteins that occur in close vicinity. It would be interesting to identify the partners and combinatorial signals that mediate the many functions of Suv4-20h proteins and H4K20 methylation.

## Figures and Tables

**Figure 1 ijms-23-04736-f001:**
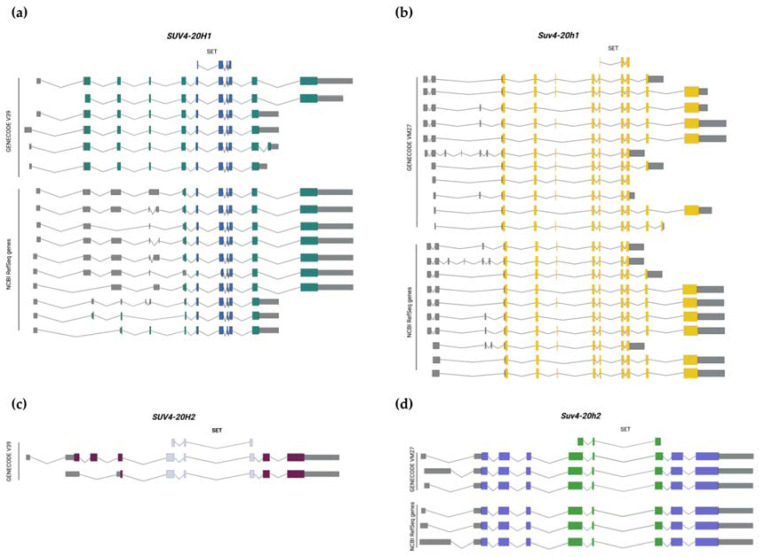
Schematic representation of the human (**a**,**c**) and murine (**b**,**d**) SUV4-20H1/Suv4-20h1 and SUV4-20H2/Suv4-20h2 transcript variants. Colored thick boxes represent coding exons. Grey thin boxes represent non-coding exons. Exons encoding for the SET domain are shown above the isoforms and are highlighted in different colors. Both GENECODE annotated isoforms and RefSeq predicted isoforms are reported (created with BioRender. Available online: https://biorender.com/ (accessed on 6 April 2022) [15]).

**Figure 2 ijms-23-04736-f002:**
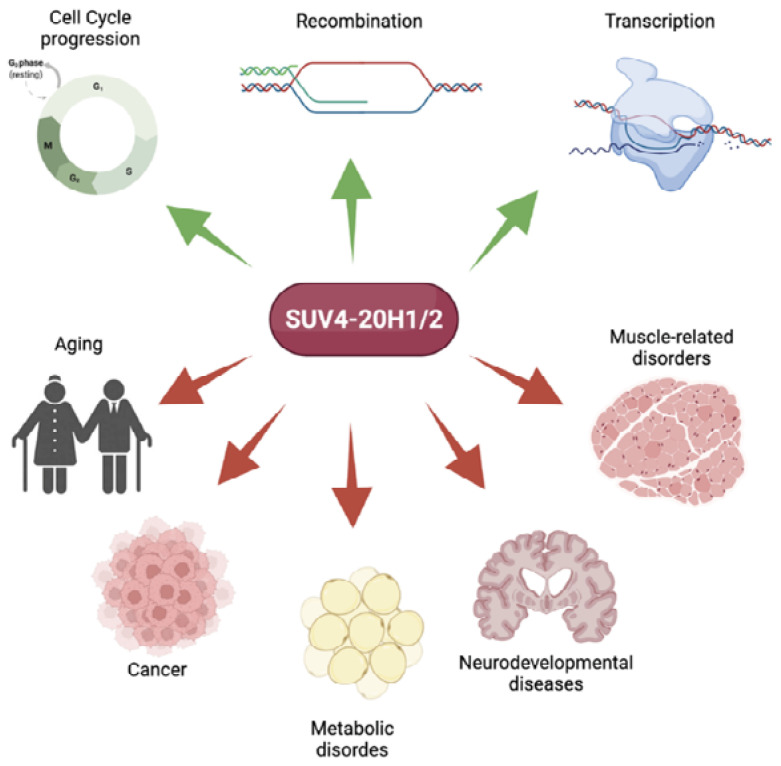
Physiological (green arrows) and pathological (red arrows) roles of SUV4-20H proteins. Both SUV4-20H1 and SUV4-20H2 are required for several physiological roles, from cell cycle progression to transcriptional regulation. The aberrant expression or genetic mutation of SUV4-20H1 and SUV4-20H2 is linked to several diseases, including aging-related diseases, neurodevelopmental and muscle-related disorders, metabolic alterations, and cancer (created with BioRender. Available online: https://biorender.com/ (accessed on 6 April 2022) [15]).

## Data Availability

Not applicable.

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
