# Peer review of "The SUV4-20H Histone Methyltransferases in Health and Disease"

_ijms, 2022, doi:10.3390/ijms23094736_

Round 1

Reviewer 1 Report

This is a concise, interesting, and well-written review manuscript about the roles of SUV4-20H histone methyltransferases for various human diseases. Their mechanisms of action and importance for disease development and progression were properly described. I suggest acceptance after minor revision:

Please describe the potential of SUV4-20A-specific histone demethylase inhibitors such as A-196 for the treatment of the described diseases, for drug development, and for the identification of the modes of action of the inhibited SUV4-20A enzymes. Give an outline of the current state of development of such inhibitors.

It was reported that EGFR modulates DNA synthesis and repair by Y72 phosphorylation of histone H4, which facilitates SUV4-20A recruitment (Chou et al., Dev. Cell. 2014, 30, 224). Please mention and discuss in the light of the report on the effects of SUV4-20A on erlotinib-resistance in cancer cells mentioned in the manuscript.

Author Response

Response to Reviewer 1 comments:

This is a concise, interesting, and well-written review manuscript about the roles of SUV4-20H histone methyltransferases for various human diseases. Their mechanisms of action and importance for disease development and progression were properly described. I suggest acceptance after minor revision:

Point 1: Please describe the potential of SUV4-20A-specific histone demethylase inhibitors such as A-196 for the treatment of the described diseases, for drug development, and for the identification of the modes of action of the inhibited SUV4-20A enzymes. Give an outline of the current state of development of such inhibitors.

Response 1: We thank the Reviewer for the positive comments and for highlighting such important missing point. In the Discussion section, we have provided an overview about the usage of A-196 inhibitor for disease treatment, also speculating about future potential developments.

Point 2: It was reported that EGFR modulates DNA synthesis and repair by Y72 phosphorylation of histone H4, which facilitates SUV4-20A recruitment (Chou et al., Dev. Cell. 2014, 30, 224). Please mention and discuss in the light of the report on the effects of SUV4-20A on erlotinib-resistance in cancer cells mentioned in the manuscript.

Response 2: As suggested by the Reviewer, we have discussed the report form Chou et al. in the Cancer section (see page 8 of the revised manuscript).

Reviewer 2 Report

This is a actually well written, comprehensive review on the SUV4-20H proteins -- histone lysine methyltransferases that modify/methylate H4K20 (histone H4 methylation on N-terminal tail lysine 20) as dimethyltransferase and trimethyltransferase. This article provides in-depth history, studies, summaries on the significance of the proteins and the field, which is a great review for both researchers who are new in the field as well as seasonal veterans.

There are no major concerns to the article. Minor suggestions is that Figure 2 labeled "Aging" over a figure of a presumably a woman who also appeared to weighted above average. Since the article discussed aging and obesity separately, I would suggest to break the icon into two (since 'Aging' and 'Obesity' do not have to be developed simultaneously), preferable use logos that are gender neutral.

Author Response

Response to Reviewer 2 comments:

Reviewer 2:

This is a actually well written, comprehensive review on the SUV4-20H proteins -- histone lysine methyltransferases that modify/methylate H4K20 (histone H4 methylation on N-terminal tail lysine 20) as dimethyltransferase and trimethyltransferase. This article provides in-depth history, studies, summaries on the significance of the proteins and the field, which is a great review for both researchers who are new in the field as well as seasonal veterans.

Point 1: There are no major concerns to the article. Minor suggestions is that Figure 2 labeled "Aging" over a figure of a presumably a woman who also appeared to weighted above average. Since the article discussed aging and obesity separately, I would suggest to break the icon into two (since 'Aging' and 'Obesity' do not have to be developed simultaneously), preferable use logos that are gender neutral.

Response 1: We thank the Reviewer for the positive comments and for bringing to our attention the misleading Figure. We have now replaced the Aging icon by using a new one depicting both male and female aged persons. Unfortunately, we were unable to find a gender neutral logo. A separate icon (bulk of white adipocytes) has been used for highlighting the involvement of Suv4-20h proteins in metabolic disorders.